# State of Mind Assessment in Relation to Adult Attachment and Text Analysis of Adult Attachment Interviews in a Sample of Patients with Anorexia Nervosa

Cristina Civilotti [1,2,†], Martina Franceschinis [2,†], Gabriella Gandino [1,*], Fabio Veglia [1], Simona Anselmetti [3], Sara Bertelli [3], Armando D'Agostino [3,4], Carolina Alberta Redaelli [3], Renata del Giudice [3,4], Rebecca Giampaolo [1], Isabel Fernandez [5], Sarah Finzi [1], Alessia Celeghin [1], Edoardo Donarelli [1] and Giulia Di Fini [1]

1    Department of Psychology, University of Turin, 10124 Turin, Italy
2    Faculty of Educational Science, Salesian University Institute (IUSTO), 10155 Turin, Italy
3    Department of Mental Health, ASST Santi Paolo e Carlo, 20142 Milan, Italy
4    Department of Health Sciences, University of Milan, 20142 Milan, Italy
5    Center of Research and Studies in Psychotraumatology (CRSP), 27045 Milan, Italy
*    Correspondence: gabriella.gandino@unito.it; Tel.: +39-338-5818083
†    These authors contributed equally to this work.

**Abstract:** Background: Attachment theory represents one of the most important references for the study of the development of an individual throughout their life cycle and provides the clinician with a profound key for the purposes of understanding the suffering that underlies severe psychopathologies such as eating disorders. As such, we conducted a cross-sectional study with a mixed-methods analysis on a sample of 32 young women with anorexia nervosa (AN); this study was embedded in the utilized theoretical framework with the following aims: 1. to evaluate the state of mind (SoM) in relation to adult attachment, assuming a prevalence of the dismissing (DS) SoM and 2. to analyze the linguistic attachment profile emerging from the transcripts of the AAIs. Methods: Interviews were transcribed verbatim, coded, and analyzed using the linguistic inquiry and word count (LIWC) method. Results: The results were observed to be consistent with the referenced literature. The prevalence of a DS SoM (68.75%) is observed in the study sample, whereas the results of the lexical analysis of the stories deviate from expectations. Notably, the lexical results indicate the coexistence of the dismissing and entangled aspects at the representational level. Conclusions: The study results suggest a high level of specificity in the emotional functioning of patients with AN, with a focusing on a pervasive control of emotions that is well illustrated by the avoidant/ambivalent (A/C) strategy described in Crittenden's dynamic–maturational model. These findings and considerations have important implications for clinical work and treatment, which we believe must be structured on the basis of starting from a reappraisal of emotional content.

**Keywords:** AAI; attachment; eating disorders; anorexia nervosa; textual analysis; LIWC

## 1. Introduction

Anorexia nervosa (AN) is described in the DSM-5 according to the following diagnostic criteria [1]: a. restriction of caloric intake relative to need, resulting in significantly low body weight relative to age, gender, developmental history, and physical health; b. intense fear of weight gain; and c. change in the way weight or body shape is experienced by the person, with an excessive impact of weight or body shape on self-esteem or a persistent lack of recognition of the severity of the current underweight condition.

In Europe, the incidence of AN is 100–200 cases per 100,000 young women within the risk age bracket (i.e., 15–19 years), with a prevalence of between 1 and 4% between the ages of 12 and 18 years [2]. AN typically occurs during adolescence and early adulthood, although cases are also reported with early onset (before puberty) or late onset (after age

40), and is often associated with a stressful event and tends to be common—particularly in the female population. It is estimated that the average lifetime prevalence of AN is 1.4% in women and 0.2% in men [3]. Although most patients with AN (66.8%) can be considered clinically cured within 5 years of onset [4], approximately 10–20% of patients develop a chronic condition [1], and relapse rates remain very high [5,6]. Eating behaviors in patients with AN are not only responsible for the development of dangerous medical conditions but also affect their quality of life and lead to functional limitations, especially in the area of interpersonal relationships. Overall, such a clinical picture explains the high crude mortality rate of AN, which is 5% according to the DSM-5 [1] but estimated at 4% according to the most recent literature review. According to Galmiche and colleagues [3], mortality is most often the result of medical complications related to the disorder itself or suicide. Therefore, most recognized institutes for the study and monitoring of this eating disorder phenomenon support the importance of research in order to develop effective intervention programs [7,8]. Analysis of the totality of the interacting factors of a biological, psychological, and sociocultural nature that contribute to the occurrence and maintenance of all eating disorders represents an essential prerequisite for a basic understanding of the complexity of these disorders and for improved knowledge of the specificities of each factor. Therefore, the aim of this study is to draw attention to one of the most salient and clinically significant aspects of AN—which is thought as one of the possible perspectives from which to view a complex phenomenon such as AN—as well as to the metacognitive content typically associated with the disorder and to the specific underlying psychological factors that, in interaction with a system of maladaptive beliefs, trigger and maintain the pathology itself.

### 1.1. The Problems of Emotion Regulation in Patients with AN

In recent years, a large number of studies have addressed the detailed description of the multifactorial etiopathogenesis of eating disorders and the clarification of the factors contributing to their maintenance, thereby highlighting a complex system of interaction between sociocultural and environmental, temperamental, and psychological factors.

Environmental risk factors include the idealization and glorification of the female slimness ideal promoted by prevailing cultural models, as well as the presence of traumatic life events [9]. However, there are individual differences related to other environmental, temperamental, psychological, and genetic factors that attenuate susceptibility to internalizing cultural messages and place the individual at lower or higher risk of developing EDs [10]. Emotional experiences have been recognized as a factor associated with AN since the first clinical observations of the disorder. It is clear that there is a specific relationship between the emotional experience and behavioral expression of AN, which is the result of specific emotional regulation strategies based on avoidance [11]. The goal of avoiding emotional experiences is pursued through certain predictable and controllable behaviors. For example, hyperfocusing on food, diet, weight, and body shape; excessive exercise and elimination behaviors; and adopting metacognitive processes, such as rumination, are some of the behaviors that affected individuals use in an attempt to divert attention away from emotional content [12,13]. In addition, physiological changes resulting from starvation and depletion contribute to emotional blunting [14,15], as does the establishment of dysfunctional ways of coping with stress [15]. Moreover, the increasingly impaired quality of interpersonal relationships contributes to the chronification of the disorder [16]. The dysfunctional regulation of emotions not only plays an essential role in the development and maintenance of AN but also represents the most robust mediating mechanism involved in the relationship between insecure attachment and eating symptoms [17–23]; in other words, this means that the close relationship between insecure attachment and AN can be explained in light of emotional dysregulation. Thus, according to this line of research, structural difficulty at the level of emotional functioning underlies the consolidation of maladaptive (hyperactivating and deactivating) emotional regulation strategies. These strategies stem from an early insecure attachment relationship [24–26], which manifests as

eating symptoms and certain typical behaviors, which represent an attempt to counteract the discomfort stemming from the emotional confusion that characterizes the state of mind of patients with AN [27,28]. According to the maintenance model of AN as described in detail by Oldershaw and colleagues [29], the maladaptive patterns that typically describe the experiences of individuals with AN result from some aspects that are related to early life experiences such as attachment quality, specific genetic factors, personality and temperamental traits (in particular the tendency toward perfection, harm avoidance, and persistence), and neurocognitive and sociocognitive difficulties (such as instances of hyperfocusing on details and attentional bias toward threats). These factors, defined as the background, contribute to the emergence of low self-awareness and a weak self-concept, as well as to the entrenchment of maladaptive beliefs regarding emotions that are considered unacceptable, dangerous, and uncontrollable or—in regard to their expression—that which is considered likely to trigger criticism and rejection. In this context, a deeply negative value is attributed to the so-called "primary", i.e., functional and appropriate, emotions that determine the patient's active avoidance and promotes the affirmation of the so-called "secondary" emotions (e.g., shame) on which the entire emotional experience of the individual then depends [30]. Simultaneously with the avoidance of the adaptive emotions, an enhancement of the secondary emotions is observed as an aspect that may explain, at least in part, the significantly higher levels of feelings of disgust, shame, and guilt observed in the clinical population with AN compared to control groups [31]. In the maintenance cycle, the essentially maladaptive and dysregulated emotional experience also contributes to the difficulties of patients with AN in developing adaptive strategies for improved emotional regulation [32]. During each stage of the emotional process, such patients tend to prefer maladaptive strategies based on avoidance, rumination, and suppression of the emotional response.

Finally, important evidence in the literature suggests that patients with AN have difficulties with respect to emotional self-awareness [33,34]. These difficulties are the result of the presence of marked alexithymia, i.e., a deficit in the process of identifying, expressing, and describing emotions, which has been widely associated with AN in the literature, as well as a lack of emotional clarity [35]. Although the active avoidance of one's emotional world is observed, in addition to alexithymic processes, patients with AN report significant abnormalities ranging from experiencing high levels of specific negative emotions (mainly shame and disgust) to dependence and incompetence, feelings of isolation and helplessness, and negative beliefs about feeling and expressing emotions [36].

### 1.2. Life History and Traumatic Experiences: The Role of the Attachment System

Initially, the clinical literature on traumatic experiences and eating disorders focused on childhood sexual abuse because positive and consistent correlations between sexual abuse and bulimia nervosa, as well as between sexual abuse and binge eating disorder [37] have been reported. Recently, research has focused on other forms of child abuse, particularly physical abuse and emotional abuse, which, similar to sexual trauma, can lead to significant consequences in adulthood. Significant associations have been found between physical abuse and every form of eating disorder, including AN, whereas emotional abuse has been positively correlated only with bulimia nervosa and binge eating disorder to date [38]. Sexual, physical, and emotional abuse (including physical and emotional neglect) experienced in childhood falls into a broader category of what are known as adverse childhood experiences, a term used to describe all experiences or life events that interfere with normal development during childhood and functioning and that can potentially lead to adverse outcomes [39]. In the life history of a person with traumatic experiences, dysfunctional eating behaviors appear to represent an attempt (albeit a maladaptive one) to cope with unbearable memories and disorganizing emotions, both of which have their roots in adverse experiences.

From a classification perspective, one of the most important transdiagnostic features that unites all eating disorders is attachment insecurity [40]. This statement is supported by

a large body of evidence provided in the last two decades of research and which has been reviewed in the recent literature. There are reports of solid data regarding the existence of an increased prevalence of an insecure mental state in clinical samples with eating disorders compared to healthy controls [18,40–44]. Specifically, attachment insecurity has a prevalence ranging from 70% in eating disorder patients [45,46] to 95% [47] and 100% [48]; it has also been considered an important risk factor for the occurrence of dysfunctional eating behaviors in the nonclinical population [19,49]. Traumatic life experiences, especially when experienced early and repeatedly over time, considerably increase this risk and play a major role in both the development and maintenance of dysfunctional circuits [50]. These experiences are rooted in the childhood strategies of insecure attachment (avoidant pattern (A), ambivalent pattern (C), and disorganized pattern (D)) and then in adult SoM (dismissing (DS), entangled (E), unresolved (U), or cannot classify (CC)) [51,52].

In general, the distancing strategies typical of the dismissing SoM (DS) [52–54] are characterized by the interviewee's tendency to provide a positive, idealized account of their experiences and attachment relationships but without providing episodes or memories to support it, which entails the production of untrue speech and thus a violation of the highest quality. The subject appears to be strongly uncomfortable when dealing with attachment issues and with memories that are typically avoided (e.g., no mention of pain or distress), devalued (e.g., derogatory comments about vulnerability), idealized (e.g., positive evaluation of negative experiences because it made them strong), minimized, or normalized (e.g., tendency to describe the relationship as normal). By describing themself as independent, strong, or normal and the relationship with their parents as normal, perfect, memoryless, or worthless, the respondent implicitly distances themself from the issues and needs of attachment. Overall, the story seems abstract and lacks the emotional complexity and depth that would typically characterize a balanced, fair, and objective discourse.

On the contrary, the demonstrated hyperactivating strategies are typical of an entangled (E) SoM [53–55], whereby the narrative is essentially incoherent. The tendency to produce a vague and convoluted account of attachment experiences represents an emblematic violation of the world's utmost conversational capacity. The interviewee is shown to be psychologically enmeshed in attachment relationships at the time of the interview and is still engaged in revising and reconstructing them in an adult mental representation—even if it is still entangled. The narrative also tends to be unreasonably rambling, for example, including episodes from the present in telling the story of events unrelated to the interview subjects, and/or in describing relationships with persons other than the attachment figures. The subject appears to be heavily involved in recalling their attachment history to the point of being passive, fleeting, obscure, or, on the contrary, full of strong and confused feelings (typically anger toward the parents).

An unresolved (U) SoM [53,55–57] (it must be noted that this category joins the three main categories discussed previously) is characterized by linguistic markers showing a lack of elaboration and integration in the memory of trauma or loss in the life course. In the portion of the interview in which the person reports grief or trauma suffered, typical disruptions in thought monitoring, disruptions in language monitoring, and/or extreme behavioral responses are observed. In 2009, Bakermans-Kranenburg and van Ijzendoorn [58] conducted an important meta-analysis of more than two hundred studies conducted over the past twenty-five years on the presentation of attachment in adults. In the nonclinical population, their study revealed that 50% of people have an F state of mind, 24% have a DS state of mind, 9% have an E state of mind, and 16% have an unresolved state of mind, which is observed in most cases (about 10%) among the entangled and, to a much lesser extent (about 2%), among the free. In the clinical population, the distribution changes considerably, with 21% of subjects possessing an F state of mind, 23% with a DS state of mind, 13% possessing an E state of mind, and 43% with an unresolved state of mind.

Based on studies conducted in families at high psychosocial risk, Crittenden modified and extended the most widely accepted attachment models to date and developed a com-

plex biopsychosocial model of attachment and adjustment called the dynamic–maturational model (DMM) [59–63].

From a dynamic–maturational perspective, the attachment models described by Ainsworth and colleagues [64,65] are understood as mental and behavioral strategies for the purposes of self-protection and to find a reproductive partner [66]. However, the central goal of Crittenden's work on attachment was to elaborate a model of these strategies that takes into account not only early attachment experiences and the relationship with attachment figures from which they are constituted in childhood but also all those maturational and contextual elements that change over the life course and the interplay of which helps to structure behavioral strategies [67]. Specifically, the DMM entails the hypothesis that as physical and neurobiological maturation makes new and more complex mental and behavioral processes possible, changes in context provide opportunities for these processes to take hold. The development of particular organization of self-protective behavior reflects the strategies that most effectively identify, prevent, and protect the self from the dangers of particular contexts while promoting the exploration of other aspects of life. Protective strategies not only describe interpersonal behavior but also provide a functional system for diagnosis of psychopathology [66]. In this sense, the dynamic–maturation perspective offers an interesting key to interpreting psychopathology (including, in our view, AN) as the result of the selection of distorted and dysfunctional strategies at each branching point of development. The classification system for attachment proposed in the dynamic–maturational model excludes the disorganized pattern in the child (D) [68] and replaces it with a pattern defined as "avoidant/ambivalent" (A/C). This attachment configuration is associated by Crittenden [69] with behavioral or psychological problems in the child or parent and with family conditions of neglect, maltreatment, or abuse [70].

In the case of EDs, the scientific literature converges on the hypothesis that an insecure organization of state of mind is to be expected in relation to attachment [71]. However, there is no agreement in the literature about which is the most prevalent SoM. For example, Eggert, Levendosky, and Klump [72]; Friedberg and Lyddon [73]; Salzman [74]; Tereno and colleagues [75]; and Troisi and colleagues [76] all reported a high prevalence of SoM E; on the other hand, a significant higher number of papers support the hypothesis that SoM DS is most prevalent in subjects with AN [40,46–48,77–82].

In summary, examining the attachment dynamics in the life histories of ED patients and understanding the quality of their SoM are essential activities in the clinical setting, not only for the formulation of a diagnostic hypothesis but also to elucidate useful information with respect to what these aspects of the individual's functioning suggest about the therapeutic relationship, the outcome of treatment, the possible responses to separations during the therapeutic process, and the risk of dropping out [50,83,84]. Knowledge of the regulatory systems for coping with vulnerability learned in childhood, in interaction with attachment figures, makes it possible to maintain an attitude of emotional closeness appropriate to the patient, in addition to aiding in development of the therapeutic relationship and helping the patient develop new strategies to cope with stressful situations. Therefore, in order to shed light on the mechanisms related to SoMs, the aim of this research is not only to assess the prevalent SoMs in a sample of patients with AN but also to investigate their linguistic characteristics by tracing common and recurrent aspects, thus enabling the description of the linguistic profile and deepening the level of analysis. Specifically, the two main objectives of this study are as follows:

1. To evaluate the SoM in relation to adult attachment, assuming a prevalence of the DS SoM; and
2. To analyze the linguistic attachment profile emerging from the transcripts of the AAIs.

## 2. Materials and Methods

The presented study corresponds to the preliminary phase of a larger clinical trial entitled "EEG cortical connectivity in Anorexia Nervosa during Eye Movement Desensitization and Reprocessing (EMDR) treatment" conducted at an eating disorders clinic in a hospital

in the city of Milan (Italy). The research protocol was approved by the Milan Area 1 Ethics Committee on 26 January 2018 (Prot. N. 3415/2018; Reg. Sperim. N. 2017/ST/003).

### 2.1. Procedure

At the time of recruitment, all participants were informed of the objectives of the experimental design and the procedures used in the different phases of the protocol.

The main instrument used in this research was the Adult Attachment Interview (AAI), which was conducted in Italian according to the rules of the manual by George, Kaplan, and Main [85], guaranteeing its validity and reliability. Participants were interviewed individually in the morning in a quiet and comfortable room in the Eating Disorders Outpatient Clinic of San Paolo Hospital in Milan by clinical psychologists who were trained and qualified to use the AAI in research. With the participants' consent, the interviews were recorded and transcribed verbatim according to Main's [86] transcription rules. Transcripts were analyzed, codified, and scored according to the standard method described by George, Kaplan, and Main [85], which made it possible to assign each transcript to one of the classes of adult attachment states of mind and to therefore identify the predominant SoM in the clinical sample.

The transcripts of the interviews conducted according to the AAI protocol were then analyzed using the LIWC method described by Pennebaker and colleagues [87]. No segmentation was performed; instead, the entire transcripts were used, whereas previously, they were edited according to Boyd and colleagues' [88] instructions for editing texts. In particular, filler words used in spoken, everyday language were eliminated or modified so that these expressions were not erroneously counted among the linguistic categories. In addition, owing to the specificity of the texts to be analyzed, modifications suggested by Pennebaker specifically for AAI were made to the transcripts to facilitate their processing by the software (personal communication, 2012). In particular, the adaptation of the transcripts consisted of eliminating all of the text of the interview (i.e., the handover and questions of the AAI transcript) and the interviewer's comments and replacing ellipses and double lines (graphic signals indicating pauses within the AAI) with single fixed points. The transcripts, thus adapted, were then input to the LIWC software, which analyzed the content in terms of the distribution of word frequency, enabling the definition of the linguistic profile of each interviewee and the description of the narrative style of the clinical group in question.

### 2.2. Materials

The Adult Attachment Interview (AAI) [85] is a semi-structured interview that takes approximately one hour to complete and was developed in order to assess an adult person with respect to attachment. The interview consists of twenty questions that require the interviewee to reflect on memories related to attachment (up until the age of 12 years) while maintaining a coherent and collaborative discourse with the interviewer. The complexity of the questions and the emotional significance of the issues raised explain the potential ability of this interview, highlighted by the authors to "surprise the unconscious". The interview is recorded and transcribed verbatim in its entirety at a later date, thereby taking into account any disturbances or interruptions and reproducing the comments of the interviewee and the interviewer [53].

The interview evaluation is based not only on the analysis of the attachment experiences per se but, more importantly, on the analysis of the narrative ways in which the interviewee describes and reflects on these experiences. Thus, the evaluation is conducted at two levels of analysis: the level of the content of the speech and the level of the form of the speech. According to Siegel [89], it can be argued that mental representations related to attachment influence an individual's attentional and speech processes and, consequently, the form of speech. Main and Goldwyn [54] identified three main "organized" categories for classification of an adult state of mind with respect to attachment: the free state of mind (F), the dismissing state of mind (DS), and the entangled state of mind (E). This classification system was later expanded to include the additional category of unresolved

(U) [56,57]. Finally, a fifth disorganized category was identified, termed cannot classify (CC) [57,90], which is rare and typically assigned to highly inconsistent transcripts.

### 2.3. Textual Analysis

Narrative also represents an area of considerable research interest, in which various methods of qualitative investigation have been developed that make it possible to evaluate the meaning attributed to a given event by individuals with particular characteristics or by social and cultural groups. AAI transcripts provide sufficient clinical material for the purpose of performing qualitative assessments [91]. The application of a narrative approach in this research area is based on the assumption that mental representations related to attachment influence the individual's attentional and linguistic processes and, consequently, the form of speech [89]. By analyzing transcripts, it is also possible to examine the knowledge structures that the individual uses [92] and thus identify the narrative schemas that they typically employ in order to make sense of their lives and organize their views of themselves, others, and the world [93].

The study of linguistic profiles using the linguistic inquiry word count (LIWC) method is a quantitative and computer-based approach to language analysis whereby the words present in a text sample are counted and then classified into categories defined a priori by linguistics [88,94]. The LIWC method is based on the theoretical assumption that the words people use in daily life have considerable psychological value [88]. Words have historically been considered by the social sciences as carriers of beliefs, emotions, habits of thought, lived experiences, social relationships, and personality traits [95–98].

Based on the promising hypothesis regarding how individuals translate thoughts and feelings related to traumatic or stressful experiences into language can improve mental and physical health. Francis and Pennebaker conducted on of the first studies [99] to investigate and confirm the clinical value and therapeutic potential of expressive writing [100–102]. The original LIWC application has been progressively updated to incorporate more extensive dictionaries and a more sophisticated software design (LIWC2001 [103], LIWC2007 [104], and LIWC2015 [105]), culminating in the most recent version of the program, LIWC-22 87. In this study, linguistic analysis of AAI transcripts was performed using LIWC-22 in its Italian-language version (which was originally developed by Agosti and Rellini [106]).

LIWC-22 is a computer-based text analysis program that uses an internal dictionary consisting of more than 12,000 words, word stems, and frequently used phrases, each of which belongs to one or more linguistic categories called "sub-dictionaries" used to capture and assess specific psychological dimensions (emotional, cognitive, and interpersonal), as well as psychosocial constructs [88]. The LIWC-22 consists of 117 predefined categories with excellent internal consistency and generally improved psychometric properties when compared to previous versions of the instrument, which were already characterized by good reliability, external validity, content validity, and predictive validity [103,107,108]. LIWC-22 performs an analysis of the text entered into the software, which consists of counting the number of words for each of the predefined linguistic categories. After processing the text according to this criterion, the program calculates a proportional value for each category by converting the raw word count to a percentage based on the total number of words used in the text, thus obtaining data on the frequency of each category relative to the total number. The scores for the LIWC categories do not indicate how specific words are used in the text but how frequently they occur relative to the total number of words [102]. The LIWC method demonstrates the ability to identify linguistic patterns associated with a variety of dimensions also associated with AAI classifications [53], including interpersonal relationships [109,110], the emotional domain [107,111,112], attentional focus (e.g., focus on self or on others) [113–115], and cognitive styles [107,116]. In addition, the LIWC method has demonstrated the ability to identify significant linguistic patterns associated with individual differences. This instrument has been used effectively to distinguish certain groups of individuals from others based on word usage [117,118]. Thus, to this day, the

LIWC method is considered a valid tool for distinguishing between mental disorders and, consequently, distinguishing between clinical groups [94,119].

The AAI attachment groups exhibit typical features in the formal quality of language, particularly in terms of fluency and coherence [120], and therefore a specificity in linguistic style associated with the SoM typology. Thus, the further hypothesis on which this study is based is that these groups also differ from one another in terms of the frequency of their use of psychologically significant words in terms of the quality of their representation of commitment. In particular, given the expected higher prevalence of a dismissing SoM among patients in the sample, in the present study, we expected a similar distribution of frequencies of psychologically significant words, as demonstrated for this attachment group by Cassidy, Sherman, and Jones [121] (and as recently replicated by Waters and colleagues [122]). In their respective studies, the authors outlined a linguistic profile typical of dismissing individuals, the features of which elicit the formal qualities typical of the stories of such individuals described by the AAI coding system (e.g., violations of Grice's maxims of quality and quantity). In particular, Waters and colleagues' [122] LIWC-based analyses of AAI transcripts reveal a characteristic microstructure of dismissing narratives that is characterized overall by fewer words than for all other subjects [121,123]. Moreover, they are characterized by a lower frequency of words related to the emotional domain [121], particularly those related to negative emotions [124] and conjunctions and prepositions or linking words [121], as well as an increased frequency of negations [121] and words in the present tense [125]. Given the characteristics of the studied sample, the present study is expected to replicate this distribution of word frequencies.

*2.4. Statistical Analysis*

The SPSS (Statistical Package for the Social Sciences), version 27.0 [126], was used to analyze the data collected in this study.

Moreover, descriptive statistics were used to analyze the sample of patients with AN in terms of sociodemographic characteristics.

To analyze the data obtained by the LIWC-22 software in relation to the linguistic characteristics of the AAI transcripts of the sample and, in particular, to describe the linguistic profile of the clinical group in question, a *t*-test was performed with a significance level of $\alpha = 0.05$. As the research design of the present study did not include a control group, the results were compared with those reported in the literature. The sample chosen for comparison was obtained from a study by Junghaenel and colleagues' [94] in the control group. This group consisted of 10 female and 7 male volunteers from the general population. Their ages ranged from 18 to 58 years (mean = 42.00 and SD = 14.28). The educational levels of this group were as follows: high school diploma, n = 4.24%; college degree, n = 7.41%; and master's degree, n = 6.35%.

A study conducted by Junghaenel and colleagues [93] focused on the analysis of eight specific higher-level linguistic categories (i.e., personal pronouns, positive emotions, negative emotions, cognitive processes, relativity, sensory/perception, social processes, and physical states/functions), utilizing the LIWC2015 version [105]. As we only collected the necessary data for comparison with another sample (and therefore applied a *t*-test), the focus of this study is the analysis of the same linguistic categories. A high correlation was demonstrated between the LIWC2015 and LIWC-22 versions for each of the eight selected categories [88].

*2.5. Sample*

Patients were recruited from an outpatient clinic in a day hospital (i.e., not an inpatient facility) of a hospital in a large city in northern Italy.

The inclusion criteria were as follows: female, aged between 15 and 25 years, and a recent diagnosis of AN according to DSM-5 criteria [1]. The exclusion criteria for the larger research project were an inability to speak or read Italian; general medical conditions affecting eating habits, including metabolic disorders; perinatal trauma or severe

neurological disorders; severe psychiatric comorbidities (other than personality disorders); psychotherapy of any kind in the past year; and history of EMDR or CBT psychotherapy.

All patients were asked to sign an informed consent form for the study, and there were no cases of refusal to participate. Based on these criteria, a study sample was formed that consisted of 32 female subjects (N = 32) between the ages of 16 and 24 years (M = 18.59 and SD = 2.67) who met the DSM-5 diagnostic criteria related to AN (APA, 2013). Regarding the sociodemographic characteristics of the sample, in terms of ethnicity, most of the study participants were of Italian origin (84.4%), and only four participants had another origin, namely Hungarian, Tunisian, Peruvian, and Argentinian/Sinhalese. In terms of family composition, only one person was an only child (3.13%), 21 patients had only one brother/sister (65.6%), 9 patients had two (28.13%), and only 1 subject had three (3.13%) brothers/sisters. Regarding the marital status of the parents, 8 patients (25%) reported that their parents lived separately, whereas 24 patients (75%) did not report divorce or separation within the family.

Regarding the clinical characteristics of the sample, the mean age for the onset of AN was 16.71 years (SD = 2.43). A total of 16 (50%) patients were diagnosed with a restricting type of AN according to DSM-5. This was broken down as 7 (21.88%) with binge eating/purging type, 1 in partial remission (3.13%), and 8 (25%) who did not report during the questionnaire administration. Only 4 (12.5%) reported experiencing regular menstruation (however, one of them was using anticontraceptive medication).

## 3. Results

### 3.1. AAI Classification

The results show the following distribution of SoMs among the study participants. Among 32 subjects, 22 were classified as dismissing (DS, 68.75%), 7 as entangled (E, 21.88%), and 3 as free (F, 9.38%). We also identified an index of failure to cope with a loss or trauma (which affected the quality of thought processes and, consequently, the mental state) in two of the three cases, whereby an analysis of the representations of attachment experiences yielded a supposed picture of structural security. Lastly, 15 of the 32 subjects presented with an unresolved state of mind (U, 46.88%); in particular an unresolved state of mind was observed in 10 of the dismissing subjects (U/DS, 31.25%), in 3 of the entangled subjects (U/E, 9.38%), and in 2 of the secure subjects (U/F, 6.25%).

### 3.2. LIWC Categories

Through the application of the software, a list of the LIWC-22 linguistic categories in conjunction with their frequency of use in each of the 32 transcripts in question was provided, which enabled the creation of a linguistic profile for each subject and, based on comparison with a control group, the elaboration of a reflection on the linguistic profile of the entire clinical group. Table 1 shows the descriptive statistics for the study group (AN patients) and for the control group (CG), which were as described by Junghaenel and colleagues [94] in relation to the eight selected LIWC-22 language categories, as well as the data resulting from the application of the *t*-test performed at a significance level of $\alpha = 0.05$.

As shown in Table 1, there is a significant difference between the linguistic structure of the texts of the sample used for comparison and that of the clinical group, thereby showing a significant peculiarity, especially in the expression of positive and negative emotions and at the level of cognitive and perceptual processes. In the area of emotions, there is a tendency toward an overall reduction in expression of emotions. In particular, there is a lower expression of positive emotions (optimism and positive feelings) and a greater tendency to express sadness. At the cognitive level, given the increased use of terms that reflect certain thought processes, particularly causality and certainty, there is a marked lack of inhibition expressions. At the perceptual level, a marked decrease in expressions related to inclusion is accompanied by an increased frequency of exclusion expressions. Thus, at the temporal level, subjects show a strong focus on the past, a lack of attention to the present, and a lack of focus on the future. When referring to interpersonal relationships—which is

of particular relevance—a reduced reference to others and to what concerns the individual was observed, often accompanied by recurrent references to the family.

**Table 1.** The means and standard deviations of the LIWC-22 categories with significant values related to differences between AN patients and the control group.

| Higher-Level Linguistic Categories | Linguistic Sub-Categories | AN Patients | | Control Group | | t | p | df |
|---|---|---|---|---|---|---|---|---|
| | | Mean | SD | Mean | SD | | | |
| Personal Pronouns | | | | | | | | |
| | I | 5.91 | 1.24 | 10.14 | 4.03 | 5.5100 | <0.0001 *** | 47 |
| | We | 0.49 | 0.19 | 0.57 | 1.66 | 0.2718 | 0.7870 | 47 |
| | I–we | 3.20 | 2.87 | 10.71 | 3.97 | 7.6149 | <0.0001 *** | 47 |
| | You | 0.03 | 0.04 | 0.43 | 0.81 | 2.8135 | 0.0071 ** | 47 |
| | He/she–they | 0.21 | 0.25 | 0.99 | 1.13 | 3.7673 | 0.0005 *** | 47 |
| Positive Emotions | | | | | | | | |
| | Optimism | 0.42 | 0.21 | 1.29 | 0.63 | 7.1537 | <0.0001 *** | 47 |
| | Positive feelings | 1.72 | 0.55 | 2.55 | 2.52 | 2.4768 | 0.0169 * | 47 |
| Negative Emotions | | | | | | | | |
| | Anxiety | 0.34 | 0.26 | 0.40 | 0.61 | 0.4831 | 0.6313 | 47 |
| | Anger | 0.43 | 0.22 | 0.51 | 0.65 | 0.6358 | 0.5280 | 47 |
| | Sadness | 0.66 | 0.27 | 0.24 | 0.52 | 3.7383 | 0.0005 *** | 47 |
| Cognitive Processes | | | | | | | | |
| | Insight | 1.79 | 0.47 | 1.95 | 1.43 | 0.5810 | 0.5640 | 47 |
| | Causation | 1.45 | 0.39 | 0.67 | 0.60 | 5.5051 | <0.0001 *** | 47 |
| | Discrepancy | 1.46 | 0.38 | 1.56 | 1.53 | 0.3528 | 0.7258 | 47 |
| | Tentative | 2.75 | 0.79 | 2.43 | 1.80 | 0.8664 | 0.3907 | 47 |
| | Certainty | 1.93 | 0.41 | 1.25 | 1.05 | 3.2494 | 0.0021 ** | 47 |
| | Inhibition | 0.24 | 0.10 | 1.06 | 0.73 | 6.3012 | <0.0001 *** | 47 |
| Relativity | | | | | | | | |
| | Inclusion | 1.80 | 0.61 | 7.86 | 2.77 | 11.9449 | <0.0001 *** | 47 |
| | Exclusion | 5.09 | 0.80 | 3.13 | 1.93 | 5.0233 | <0.0001 *** | 47 |
| | Movement | 1.71 | 0.46 | 0.97 | 0.96 | 3.6622 | 0.0006 *** | 47 |
| | Past | 5.66 | 1.34 | 3.18 | 3.03 | 3.9804 | 0.0002 *** | 47 |
| | Present | 7.21 | 0.94 | 11.01 | 4.06 | 5.0873 | <0.0001 *** | 47 |
| | Future | 0.04 | 0.04 | 1.26 | 1.55 | 4.4920 | <0.0001 *** | 47 |
| Sensory/Perception | | | | | | | | |
| | Sight | 0.40 | 0.21 | 0.77 | 1.02 | 1.9914 | 0.0523 | 47 |
| | Hearing | 1.16 | 0.42 | 0.82 | 0.93 | 1.7676 | 0.0836 | 47 |
| | Sensations | 0.28 | 0.17 | 0.54 | 0.82 | 1.7397 | 0.0885 | 47 |
| Social Processes | | | | | | | | |
| | Communication | 1.86 | 0.56 | 1.53 | 1.96 | 0.8934 | 0.3762 | 47 |
| | Friends | 0.20 | 0.14 | 0.25 | 0.46 | 0.5716 | 0.5703 | 47 |
| | Family | 2.11 | 0.65 | 1.16 | 1.42 | 3.2221 | 0.0023 ** | 47 |
| | Human | 0.39 | 0.22 | 1.54 | 2.44 | 2.6706 | 0.0104 * | 47 |

**Table 1.** *Cont.*

| Higher-Level Linguistic Categories | Linguistic Sub-Categories | AN Patients | | Control Group | | | | |
|---|---|---|---|---|---|---|---|---|
| | | Mean | SD | Mean | SD | t | *p* | df |
| Physical States/Functions | | | | | | | | |
| | To eat | 0.12 | 0.07 | 0.13 | 0.25 | 0.2128 | 0.8324 | 47 |
| | To sleep | 0.12 | 0.13 | 0.02 | 0.10 | 2.7622 | 0.0082 ** | 47 |
| | Care | 0.03 | 0.04 | 0.02 | 1.00 | 0.0570 | 0.9548 | 47 |
| | Sexuality | 0.24 | 0.17 | 0.96 | 1.06 | 3.7858 | 0.0004 *** | 47 |
| | Body | 0.41 | 0.21 | 0.36 | 0.52 | 0.4787 | 0.6344 | 47 |

\* $p < 0.05$; \*\* $p < 0.01$; and \*\*\* $p < 0.001$.

## 4. Discussion

As expected, in the AN sample, the representations of early attachment experiences were characterized by a marked insecurity. Notably, the analyses of AAIs confirm the prevalence of a DS SoM in the population diagnosed with AN [40,43,46–48,71,77–82]. In terms of content, this is associated reference to denial behavior by attachment figures and neglect through active resistance to attachment needs (rejection). On a formal level, the interviewees' narratives appear mostly abstract and incoherent as a result of reasoning based on a certain psychological distance from the attachment issues evoked by the interview. The widespread tendency among the patients to give an implausibly positive and idealized account of their childhood experiences, often alternatively defined synthetically as "normal" through the use of global and general expressions, contributes to the definition of this characteristic without providing episodes or memories that support the assigned quality. The transcripts of the AN patients show a considerable difficulty in accessing childhood memories and a widespread tendency to devalue or minimize attachment experiences, which contributes to distancing of the subject from the emotional content they contain. In this sense, it can be said that the formal features of their language express an underlying and implicit distance from attachment experiences, especially when they are negative. The latter data are of particular importance because they are consistent with the hypothesis that individuals with AN, essentially characterized by a DS SoM, tend to avoid negative emotions through the mechanism of emotional disengagement. These considerations appear to be consistent with Connors' [127] detailed description of the affective and interpersonal functioning of a disengaged adult, who typically establishes relationships by devaluing or minimizing their importance in order to limit their impact on their life.

The E SoM is also represented in the study sample, albeit to a lesser extent. Overall, the transcripts classified as such give the impression of a current and considerable entanglement in the memories associated with attachment experiences and a difficulty in freeing oneself from the very intense emotions experienced in connection with them. This condition affects the quality of the patients' thought process and is therefore reflected in the formal quality of their narratives, which appear to be convoluted, lengthy, disjointed, and often vaguely interrupted, with no logical conclusion to the thoughts expressed. Consistent with the literature [128], a tendency to maintain ambivalent memories of caregivers can be discerned in the respondents' narratives. However, their relationship has led to the construction of an idealized model of the other that is balanced by a negative and devalued view of the self. Therefore, that the patients' perception of themselves as unworthy and unlovable, combined with the idealization of the other leads them to struggle for self-acceptance, which they can only obtain by way of recognition from other valued persons. Consequently, it is necessary for these individuals to maintain a high level of dependence on the one another in order to maintain and promote a positive valuation of the self. As suggested in the literature, this goal is pursued through a controlling behavioral style [129], which is evident in a close analysis of the examined transcripts. This is a fundamental point, as it seems that even in individuals classified with E SoM, which is less typical of the AN

diagnosis, the theme of control reappears, which is a particular behavioral feature that characterizes this disorder.

The number of transcripts classified as F was relatively low. These data therefore confirm the striking evidence in the literature, i.e., the fact that attachment wounds preclude the consolidation of susceptibility to eating disorders. As a result, this correlation deserves closer clinical attention.

The high frequency with which unresolved SoM classifications were assigned to the transcripts analyzed in this study is noteworthy. This finding is of considerable clinical interest—that is, when assuming that non-resolution within the patients indicates the presence of a psychological vulnerability in the respondents, given the inability to process a traumatic event. Moreover, this finding is in agreement with the literature on the pathogenic effects of early relational trauma and loss experiences that precede and foster the development of AN. Specifically, in the present study, unprocessing concerns loss experiences related to the death of a family member (mainly grandparents), whereas in a single case, the lack of processing and integration concerns an experience of childhood sexual abuse. From a formal point of view, in the analyzed unresolved transcripts, in the section dedicated to the exploration of traumatic experiences, a significant change compared to what was observed in the rest of the interview can be observed in the quality of the language, which takes on the features that have been dropped.

Regarding the second hypothesis of this study, which refers to the linguistic profile of the narratives in accordance with the DS SoM, the results differ from what was expected. The analysis of the frequency of use of psychologically significant words in the studied narratives shows a distribution that significantly differs from that highlighted in the literature for this attachment group [121,122]. The microstructure of the examined narratives exhibited features that seem to indicate an entanglement in, and current concern about, attachment-related issues, aspects that are typically observed in the literature for individuals with E SoM [121]. In particular, given the reduced expression of emotions consistent with expectations that are associated with distancing functioning, unexpected results emerge, particularly in relation to the expression of cognitive and perceptual processes. This level of analysis suggests that it is mainly the aspects of causation and certainty that organize the patients' thought processes and thus their stories, which appear to be complex and articulated from the logical point of view. Furthermore, they are noted to be free of expressions that denote inhibitions and appear to actively block the flow of thoughts. The lexical choice clearly shows the relevance of the experiences of exclusion in the life story of the patients. Their stories are peppered with expressions that denote a pronounced and pervasive focus on the past, to which they therefore seem to be psychologically and exclusively connected, to the detriment of a more functional way of thinking that equally takes into account the present and the future and that moves harmoniously in temporality. When comparing the entangled linguistic profile that emerged from the examination of AAI transcripts with the results of a similar exploratory study by Cassidy, Sherman, and Jones [121], it appears that the linguistic categories described by the authors as typical of the E SoM are the same for which meaning was found in the present study. The linguistic profile that emerges from the patients' narratives appears to indicate a representation of attachment that has the characteristics of both preoccupation and entanglement.

This unexpected and seemingly contradictory finding reflects the full complexity of thinking about attachment, the representations that become entrenched in adulthood, and the influence of these representations on mental and behavioral processes. If the original research hypothesis suggested the possibility of assessing and describing the mental state in relation to attachment of patients with AN using the criteria of the Main model, then the evidence obtained through the formal and linguistic analysis of the narratives therefore suggests the need to adopt a model that enables distinguishing of the possible nuances in the quality of the representation. Moreover, the results of this study are confirmed by the anomaly previously reported in a study by Pace, Guiducci, and Cavanna [71], who identified a high frequency of forms of linguistic passivity and expression of anger toward

the mother figure. These are qualitative features that are more likely to be associated with the passivity of discourse and involving anger scales, which are typical of the E SoM, than with the DS SoM. This seemingly contradictory finding reflects the complexity that the representation of attachment memories can take on in the adult psyche [130], which, according to the literature, is mainly of the dismissing type (DS) in the population with AN but may also have possible preoccupied traits, which deserve closer clinical attention [71].

The alternative theoretical hypothesis in the literature that is thought to explain the results and the apparent discrepancy between Hypothesis 1 and Hypothesis 2 is Critten-den's dynamic–maturational model for attachment [59,131]. In particular, considering the language configuration and the resulting linguistic profile, it can be assumed that it is more precise and accurate to assign patients to an avoidant/ambivalent attachment model (A/C). The study participants demonstrate the coexistence and alternation of aspects that characterize one and the other attachment configurations. The ambivalence underlying the functioning of A/C issues denotes a deep discomfort surrounding issues related to attachment that are neither permanently avoided nor pervasively occupy the individual thought processes but which determine the consolidation of mental strategies and essentially ambivalent behavior. In particular, A/C individuals do not avoid emotions a priori but, on the contrary, experience them in a pervasive manner, blocking the emotional content only after the fact, i.e., after they have had the emotional experience. The avoidance component of the A/C model occurs only in the organizational phase of the mental, behavioral, and linguistic processes, in which the emotional content, even if experienced, turns out to be forced. This model seems to sufficiently explain the representational and behavioral ambivalence that emerges at the linguistic level from the narratives of the patients involved in this study. These considerations have important implications for clinical work and the treatment of the disorder in question, which must involve a reinterpretation of the emotional content in the initial phase.

### 4.1. Strengths and Weaknesses of the Study

The current study has a number of strengths. First, one-on-one interviews allowed participants to talk about themselves and their experiences in their own words, revealing details in the data that would have been missed if self-reports had been used. Second, to the best of our knowledge, our study is the first study to combine AAI analysis and linguistic analysis in a sample of anorexic patients, with the analyses conducted independently by two different researchers. This method also allowed us to obtain more detailed knowledge of the participants' attachment problems.

The fact that the participants were selected from a broader research project may be a limitation of the study, as a large number of AN individuals who wished to tell their story may not have participated because they did not meet the inclusion criteria for the clinical trial. Furthermore, the use of attachment categories and the lens of trauma limited the contextualization of this study in relation to other scientific and well-validated perspectives. In addition, the study design neglected the influences of social aspects and neurobiological influences on emotional regulation and behavioral symptomatology.

For linguistic comparison, we used a control group from a nonclinical population described by Junghaenel et al. [94]. However, the two populations do not match in terms of sociodemographic characteristics; as such there may be some bias in the interpretation of the results. Finally, a larger sample would have allowed us to investigate possible differences within the group (e.g., in terms of age, disease duration, clinical characteristics of the disease, etc.).

### 4.2. Clinical Relevance of the Study

Attachment and emotional regulation, especially in relation to dealing with vulnerability, are inextricably linked. Anorexic patients are more likely to have insecure or unresolved attachment, which increases the likelihood that regulation of their emotions is disrupted. In order to treat anorexia, psychotherapists should also consider attachment-oriented clinical

strategies. There is a need for trauma-informed practice and appropriate interventions in order to address the additional traumatic experiences that patients with similar life histories may experience.

In addition, our findings suggest that participants may be more vulnerable to unresolved loss and trauma. Moreover, for the purposes of the development of therapeutic strategies for coping with loss, this information may be particularly important for therapists working with similar populations.

The explanation for the results of this study is consistent with recent and innovative research in the field of neurobiology that has been conducted on AN patients, suggesting that they exhibit general hypoactivity in brain regions that are involved in enteroceptive perception (e.g., the insula), the hypothalamus (involved in regulating the urge to eat), and the cerebellum (involved in controlling and planning movements). However, this takes place in conjunction with increased activity in emotional areas (e.g., the amygdala) and top-down activation in regions of the prefrontal cortex, as if there were a cognitive control downstream of a large emotional activation, without an adequate component to read internal states [132–135]. Most importantly, we believe it is key to emphasize the clinical implications of this work in light of the guidelines proposed by the NICE (National Institute for Health and Care Excellence), which emphasize the importance of clinical work that also integrates an elaborative component on emotional strategies. This underscores the medium-to-long-term efficacy of interventions for AN, including an integrative intervention plan in which behavioral and emotional aspects play distinct but synergistic roles (e.g., Ego State Therapy [136], EMDR [137–139], and CBT-E [140,141]).

### 4.3. Future Investigation

Considering the clinical relevance of the obtained results, we believe that they represent a reason for interest and attention from the scientific community and that this work can be considered an exploratory experience that offers significant opportunities for the development and deepening of future research. Therefore, we do not claim that the obtained findings and the interpretations offered herein are definitive; instead, they are intended to stimulate new thinking and further studies that can examine them in more detail. We expect future studies to be conducted with larger samples to enable intragroup comparisons and differentiation between, for example, variables such as the length and magnitude of symptoms, mapping systems of the activations at the cerebral level, and taking into account any comorbidities. Moreover, as this study mainly considers the perspective of attachment problems and emotional dysregulation, it would be important to integrate this perspective with other perspectives (e.g., social, biological, etc.) and with other psychological tools (e.g., systemic and psychodynamic approaches).

## 5. Conclusions

Attachment theory offers a solid and privileged perspective for the examination and understanding of the many ways in which psychological suffering can manifest itself. Given the importance of the quality of early experiences and attachment relationships in consolidating a state of vulnerability, as well as with respect to eating behavior psychopathology, which is closely related to the insecurity resulting from attachment wounds, this theoretical framework offers unique insights regarding research and clinical reflection on AN.

The results of this work indicate the coexistence of the dismissing and entangled aspects of SoM at the representational level within AN patients. These indications are of particular clinical importance, especially in the terms of their areas of application. Based on such a complex structure of emotional regulation, which only seems to be based on dismissing strategies, it is necessary to develop interventions that also take into account the relational entanglement by moving from behavioral protocols and drawing attention to also—if not mainly—the emotional aspects.

New perspectives are gradually gaining ground in the research tradition devoted to this topic. The contribution of this study is in taking aspects widely studied in the literature and developing them in a reflection that focuses specifically on narrative and, in particular, on the linguistic analysis of stories related to early childhood attachment experiences.

**Author Contributions:** Conceptualization, C.C. and G.D.F.; methodology, C.C. and M.F.; formal analysis, M.F. and G.D.F.; data curation, R.G., S.A., S.B., A.D., R.d.G., E.D. and C.A.R.; writing—original draft preparation, C.C. and M.F.; writing—review and editing, G.D.F., S.F., A.C., E.D. and G.G.; supervision, G.G., I.F. and F.V.; funding acquisition, I.F. and F.V. All authors have read and agreed to the published version of the manuscript.

**Funding:** This research was funded by Associazione per l'EMDR in Italia and EMDR Research Foundations. The APC was funded by the self-financed research of F.V., Department of Psychology, University of Turin.

**Institutional Review Board Statement:** This study was conducted in accordance with the guidelines of the Declaration of Helsinki and was approved by the Ethics Committee of Milan Area 1 (Prot. Number: 3415/2018).

**Informed Consent Statement:** Informed consent was obtained from all subjects involved in the study. Written informed consent was specifically obtained from the patients in order to publish this paper.

**Data Availability Statement:** The data presented in this study are available upon request from the corresponding author. The data are not publicly available due to privacy reasons.

**Acknowledgments:** We thank the patients who participated in this study and dedicated time and resources at a particularly sensitive moment in their lives. We also thank the professionals who made the development and implementation of the study possible at the Eating Disorders Unit of the San Paolo Hospital of Milan, Milan, Italy.

**Conflicts of Interest:** The authors declare no conflict of interest. The funders had no role in the design of the study; in the collection, analyses, or interpretation of data; in the writing of the manuscript; or in the decision to publish the results.

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
