# Peer review of "State of Mind Assessment in Relation to Adult Attachment and Text Analysis of Adult Attachment Interviews in a Sample of Patients with Anorexia Nervosa"

_ejihpe, doi:10.3390/ejihpe12120124_

Round 1

Reviewer 1 Report

I appreciate the opportunity to review the SoM assessment in relation to adult attachment and text analysis of Adult Attachment Interviews in a sample of patients with Anorexia Nervosa. I consider that it is an exhaustive, well documented, and planned work. I only have some suggestions for the authors, which could improve some aspect of the work.

They are the following:

- In the introduction, finalize the objectives section, differentiating between the general objective and better defined specific objectives, in the text there is an excessively broad objective.

- In methodology, indicate the approval code of the ethics committee, if available.

- Regarding the sample, is it possible to offer more data? For example, duration of the disorder at the time of participating in the research, type of AN, etc.

- Regarding the results... Could an analysis be made differentiating at least two age groups? It is expected that a 16-year-old patient is very different from a 24-year-old patient, both due to generational characteristics, duration of the disorder, previous treatment experiences, etc.

- In discussion-conclusions, I invite the authors to indicate possible theoretical and methodological limitations of the work, as well as possible future lines of research.

Author Response

Thank you for your thoughtful insightful comments. They allowed us to improve the manuscript. 

We added the objectives in a more specific way at the end of the introduction.

We have added the required authorization code/registration number.

Regarding the sample, we added more data related to some socio-demographical information and to the clinical characteristics.

Regarding the results, unfortunately the sample size does not allow us to do sub-analyzes, but it is an interesting suggestion and we have included it both as a constraint and for future directions in the three paragraphs at the end of the discussion (Strengths and weaknesses of the study, clinical relevance of the study, and future investigations).

Finally, we added three short paragraph at the end of the discussion.

Reviewer 2 Report

Thank you for the opportunity to review this paper.

Overall, the language in this paper is characterized by an idea about eating disorders, that focuses mainly on the lack of abilities that people with AN have. I wonder if you more clearly could state that this is one way or framework of understanding the challenges that accompany an eating disorder.  

I have some difficulties with the language, especially when the sentences get too long.

My biggest concern with this paper is the lack of coherence between the introduction/aim of the study and your conclusion, which - I believe - make the discussion and conclusion over-conclusive. There is a small sample size, and most of the arguments in the discussion are based on theoretical observations rather than research.

It´s also very difficult for me to see, how the results from this study can be of clinical significance, because it´s unclearly written in the paper.

I have some comments, that might improve the paper:

Line 86: What does DCA stand for?

Lines 92-93: Ruminations is stated twice

Lines 102-107: This sentence is very long, which makes it rather unclear

Lines 108-129: Is all the text based on Oldershaw et al.? Please add reference/s 

Lines 241-244: In what way is this relevant to the therapeutic relationship, treatment outcome and dropout?

Line 311: What do you mean by ‘AI is one of the best examples in this sense’? The best to do what?

Line 404-405: Why are experiences with psychotherapy, and history of EMDR/CBT exclusion criteria? What kind of treatment did the outpatient clinic offer, if not psychotherapy?

Line 443: There is no information on the sample used for comparison

Line 463: How do the authors know that it´s ‘untrue’?

579-582: It is difficult for me to see, how your results are in line with neurobiological research

Lines 587-590: I fail to see the clinical implications based on your results

Lines 594-597: This does not correspond with the introduction, in which you claim a relevance to the therapeutic relationship, treatment outcome and dropout

Author Response

Thank you for your thoughtful insightful comments. They allowed us to improve the manuscript. 

We have added the clarification that this is one way or framework of understanding the challenges that accompany an eating disorder, in several passages of the text (introduction, limitation, and future directions).

We have revised the text with the help of MDPI professional editing and translation service to solve this problem.

Regarding the lack of coherence between the introduction/aim of the study and the conclusion we tried to better clarify these aspects in all the manuscript.

Line 86: What does DCA stand for?

A: That was a mistake! DCA stands for the Italian version of Eatind Disorder. Even after reading and revising the article several times, the error remained! We have corrected it. We thank you for noticing it.

Lines 92-93: Ruminations is stated twice

A: We fixed it, thank you.

Lines 102-107: This sentence is very long, which makes it rather unclear

A: We have revised the text with the help of a professional editing and translation service to solve all these problems.

Lines 108-129: Is all the text based on Oldershaw et al.? Please add reference/s  

A: We added the references for each phrase. Thanks.

Lines 241-244: In what way is this relevant to the therapeutic relationship, treatment outcome and dropout?

A: We better (but briefly) explained how having in mind the SoMs of AN patients helps the therapeutic relation.

Line 311: What do you mean by ‘AI is one of the best examples in this sense’? The best to do what?

A: We better specified this sentence.

Line 404-405: Why are experiences with psychotherapy, and history of EMDR/CBT exclusion criteria? What kind of treatment did the outpatient clinic offer, if not psychotherapy?

A: Thanks. We added an explicit reference to the greater research project, as presented at the beginning of the “material and methods” sections. Exclusion criteria are necessary for the actual RCT.

Line 443: There is no information on the sample used for comparison

A: The sample selected for comparison was from an existing paper by Junghaenel and colleagues. We have provided all specifications as described in the original paper and noted in the limitations that bias may occur in the interpretation of the results because the two populations differ in their sociodemographic characteristics.

Line 463: How do the authors know that it´s ‘untrue’?

A: Thanks. The correct word was “incoherent”.

579-582: It is difficult for me to see, how your results are in line with neurobiological research

A: We better specified it in the “clinical implications” paragraph.

Lines 587-590: I fail to see the clinical implications based on your results

A.We better explained this point in the “clinical implications” paragraph.

Lines 594-597: This does not correspond with the introduction, in which you claim a relevance to the therapeutic relationship, treatment outcome and dropout

A: We better explained this point in the “clinical implications” paragraph.

Reviewer 3 Report

In this interesting work, the transcripts of the Adult Attachment Interview (AAI) of a sample of 32 young women suffering from anorexia nervosa (AN) have been studied, (i) to evaluate  the State of Mind (SoM) of the participants in relation to AAI classification as theorised by Mary Main and her collaborators, and, (ii) to analyse the linguistic profile of the participants. The Linguistic Inquiry Word Count has been used for the analysis. The underlying assumption has been that Dismissing SoM is the dominant category for AN in the AAI.

While the first item (i) above is confirmed in the study, the second item (ii) is not. The linguistic profile of the participants is more consistent with a model that allows an avoidant/ambivalent (A/C) category. The findings in fact provide evidence that, for AN at least, the Dynamic Maturational Model (DMM) of Crittenden is a better categorisation of adult attachment than that of Main et al.

The reviewer thinks that the findings of this study, albeit with a small sample size, are plausible since DMM has 22 basic types as well as mixed types and thus provides a more refined and nuanced model of adult attachment than Main’s model which only has a handful categories.

The paper is generally very well written and there are only minor changes required for the revised version.

One of my main issues is this: Many of the paragraphs in the paper are too long and should be broken to smaller ones for ease of reading.  This is not too difficult to do for the authors, but it will help the reader a lot.

Minor corrections and suggestions:

L2 Please use State of Mind rather than SoM in the title. All abbreviations should only be made after the full phrase is first provided with the abbreviated short form in brackets.

L40 for the clinic à for clinical work

L86 You use ED later on for Eating Disorder so I suggest you change DCA to “eating disorder (ED)”.

L148 neglect-experienced in childhood à neglect, experienced in childhood,

L168 The main reference for adult SoM is not correctly given in 48-52 that are cited here. In fact only 52 mentions the relevant categories.

L187 The phrase “reconstructing them in a mature mental representation” can be confused with the reorganization strategy of Crittenden for moving from an insecure to a secure attachment. Please re-phrase.

L285 an adult’s à an adult

L311 social group. and cultural à social and cultural groups.

L311 AI is à AAI is

L362-363 Please rewrite as:

higher prevalence among patients in the sample of a Dismissing SoM, the present study

à

higher prevalence of a Dismissing SoM among patients in the sample, the present study

L390-396 The number of linguistic categories is given as 7, or 8 or 9 which is confusing. While the reader expects 9 categories in the bracket (lines 391 and 392) only 7 is given and later in 396 the number is given as 8.

L422 it is considered à we consider it

L428 Is “distant” a standard term for U/Ds?

L434 Please say a few words about the control group to explain how it was set up.

L459 analyzes  à analysis

L506 unresolution à non-resolution

L513-517 The long sentence “From a formal point of view….” which contains “dropped. In U” does not make sense and should be rewritten, best as two sentences.

L583 the clinic à clinical work

L585-586  deepening. In future research à deepening in future research.

Author Response

Thank you for your thoughtful insightful comments. They allowed us to improve the manuscript. 

Regarding the paragraphs in the paper that were too long we have revised the text with the help of MDPI professional editing and translation service to solve all these problems.

Regarding SoM in the title we fixed it and thanks for pin-pointed it.

L40 for the clinic à for clinical work

A: Thanks. Accepted.

L86 You use ED later on for Eating Disorder so I suggest you change DCA to “eating disorder (ED)”. 

A: DCA stands for the Italian version of Eatind Disorder. Even after reading and revising the article several times, the error remained! We have corrected it. We thank you for noticing it.

L148 neglect-experienced in childhood à neglect, experienced in childhood, 

A: Thanks. Accepted.

L168 The main reference for adult SoM is not correctly given in 48-52 that are cited here. In fact only 52 mentions the relevant categories. 

A: Thanks. Accepted.

L187 The phrase “reconstructing them in a mature mental representation” can be confused with the reorganization strategy of Crittenden for moving from an insecure to a secure attachment. Please re-phrase. 

A: Thanks. Accepted.

L285 an adult’s à an adult

A: Thanks. Accepted.

L311 social group. and cultural à social and cultural groups. 

A: Thanks. Accepted.

L311 AI is à AAI is 

A: Thanks. Accepted.

L362-363 Please rewrite as: 

higher prevalence among patients in the sample of a Dismissing SoM, the present study

à

higher prevalence of a Dismissing SoM among patients in the sample, the present study

A: Thanks. Accepted.

L390-396 The number of linguistic categories is given as 7, or 8 or 9 which is confusing. While the reader expects 9 categories in the bracket (lines 391 and 392) only 7 is given and later in 396 the number is given as 8. 

A: Thanks. The categories are 8. We checked in all the paragraphs and put the labels exactly as were reported in the original Junghaenel’s article.

L422 it is considered à we consider it 

A: Thanks. Accepted.

L428 Is “distant” a standard term for U/Ds? 

A: Thanks. We changed with the more appropriate “dismissing”.

L434 Please say a few words about the control group to explain how it was set up. 

The sample selected for comparison was from an existing paper by Junghaenel and colleagues. We have provided all specifications as described in the original paper and noted in the limitations that bias may occur in the interpretation of the results because the two populations differ in their sociodemographic characteristics.

L506 unresolution à non-resolution 

A: Thanks. Accepted.

L513-517 The long sentence “From a formal point of view….” which contains “dropped. In U” does not make sense and should be rewritten, best as two sentences. 

A: Thanks. Accepted.

L583 the clinic à clinical work

A: Thanks. Accepted.

L585-586  deepening. In future research à deepening in future research.

A: We rephrased al the section, adding three paragraphs.

Round 2

Reviewer 2 Report

Thank you for your revised manuscript. I think your revisions has improved the manuscript substantially, and I have no comments for this version. Good luck with your further research on this important topic of improving treatment for eating disorders.